# Simultaneous Missing Value Imputation and Structure Learning with Groups

**Pablo Morales-Alvarez**[*][†]
University of Granada

**Wenbo Gong**
Microsoft Research

**Angus Lamb**[†]
G-Research

**Simon Woodhead**
Eedi

**Simon Peyton Jones**[†]
Epic Games

**Nick Pawlowski**
Microsoft Research

**Miltiadis Allamanis**[†]
Google

**Cheng Zhang**
Microsoft Research

## Abstract

Learning structures between groups of variables from data with missing values is an important task in the real world, yet difficult to solve. One typical scenario is discovering the structure among topics in the education domain to identify learning pathways. Here, the observations are student performances for questions under each topic which contain missing values. However, most existing methods focus on learning structures between a few individual variables from the complete data. In this work, we propose VISL, a novel scalable structure learning approach that can simultaneously infer structures between groups of variables under missing data and perform missing value imputations with deep learning. Particularly, we propose a generative model with a structured latent space and a graph neural network-based architecture, scaling to a large number of variables. Empirically, we conduct extensive experiments on synthetic, semi-synthetic, and real-world education data sets. We show improved performances on both imputation and structure learning accuracy compared to popular and recent approaches.

## 1 Introduction

Understanding the structural relationships among different variables provides critical insights in many real-world applications, such as medicine, economics and education [42, 62]. Thus, learning graphs from observed data, known as structure learning, has recently made remarkable progress [10, 61, 63, 64].

For many applications, variables in the data can be gathered into semantically meaningful groups, where useful insights are at group level. For example, in finance, one may be interested in how a financial situation influences different industries (i.e. groups) instead of individual companies (i.e. variables). Similarly, in education, the data can contain student responses to thousands of individual questions (i.e. variables), where each question belongs to a broader topic (i.e. groups). Again, it is insightful to find relationships between topics instead of individual questions. Moreover, real-world data such as educational data is inherently sparse since it is not feasible to ask every question to every student; the dimensions of the data in terms of the number of variables and the number of observations are very high, posing a scalability challenge. Despite the progress in structure learning, no existing method can discover group-wise relationships given large-scale partially observed data.

In this work, we present VISL (missing value imputation with structural learning), a novel approach to simultaneously tackle group-wise structure learning and missing value imputations driven by the

---

[*] Department of Statistics and Operation Research, University of Granada, Spain
[†] Work done while internship in Microsoft Research Cambridge.
Correspondence to: Cheng Zhang <Cheng.Zhang@microsoft.com>

36th Conference on Neural Information Processing Systems (NeurIPS 2022).

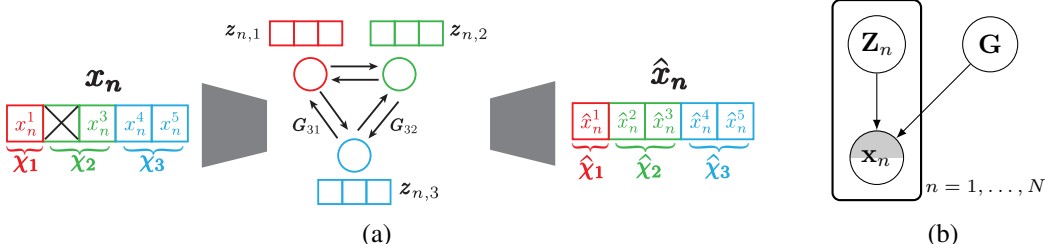

Figure 1: (a) Graphic representation of VISL. VISL is a variational auto-encoder based framework. Observations from each group are encoded into low dimensional latent variables. The structure is treated as a global latent variable. A GNN based decoder is used to decode the latent variables to observations. (b) Probabilistic graphical model for VISL, where the partial observation $\mathbf{x}$ is generated from its local latent variable $\mathbf{z}$ and the global latent variable $\mathbf{G}$ which characterizes the structures.

real-world topic relationship discovery in an education setting. This is accomplished by combining variational inference with a generative model that leverages a structured latent space and a decoder based on message-passing Graph Neural Networks (GNN) [13]. Namely, the structured latent space endows each group of variables with its latent subspace, and the interactions between the subspaces are regulated by a GNN whose behavior depends on the inferred graph from variational inference, see Fig. 1(a). VISL satisfies all the desired properties: it leverages continuous optimization of the structure learning to achieve scalability [63, 64]; the VISL formulation naturally handles missing values, and it can discover relations at different levels of granularity with pre-defined groups. Empirically, we evaluate VISL on one synthetic and two real-world problems including the aforementioned education scenario. VISL shows improved performance in both missing data imputation and structure learning accuracy compared to popular and recent approaches for each task. We worked closely with an education domain expert to evaluate the learned topic relationships, and our model has provided insightful results as recognized by the domain experts.

## 2 Model Description

In the following, we present the formulation of VISL for scalable group-wise structure learning with partial observations using a novel deep generative model based framework.

### 2.1 Problem setting

Assume a training data set $\mathbf{X} = \{\mathbf{x}_n\}_{n=1}^N$ with $\mathbf{x}_n \in \mathbb{R}^D$. The observed and missing values are denoted as $\mathbf{X}_O$ and $\mathbf{X}_U$, respectively, where we assume the data are missing completely at random (MCAR) or missing at random (MAR). In Appx. A, we explain how to handle MAR. In particular, variables can be gathered into $M$ groups, where each can be denoted as $\chi_{n,m} = [x_{n,i}]_{i \in \mathcal{I}_m}$. $\mathcal{I}_m$ containing the variable indices belonging to group $m$ (e.g., $\mathcal{I}_2 = [4, 5, 6]$ indicates group 2 includes the 4th, 5th and 6th variables). One should note that each $\mathcal{I}_m$ may have varying sizes for different $m$ (i.e. varying group sizes). Throughout the paper, we assume the group information is provided a priori. If this information is unavailable, casual representation learning can be leveraged to represent low-level signals into groups. The goal of VISL is to (i) perform missing value imputation for test samples and (ii) infer structures between groups of variables. We use the adjacency matrix $\mathbf{G} \in [0, 1]^{M \times M}$ to represent a graph, where $G_{ij} = 1$ or $0$ indicates whether there is a directed edge from $i-$th to $j-$th group or not. In the context of the education domain, the above formulation can be rephrased as follows: variable $\mathbf{x}_n$ containing the student's responses to a set of questions. $x_{i,j} = 1$ represents student $i$ has answered question $j$ correctly. Groups can be defined as the topic associated with each question. $\mathcal{I}_m$ contains the question IDs that belong to the same topic, and $\chi_m$ represents a group of responses related to that topic. Clearly, not all students can answer every question. Thus, $\mathbf{X}_O$, $\mathbf{X}_U$ represent the existing responses and un-answered questions, respectively. The goal of VISL is to (i) predict students' responses to un-answered questions, which by itself is important in the education domain [56, 57], and (ii) discover the relationships between topics, which can help education experts optimize the learning experience and the curriculum. For structure learning, we adopt a Bayesian approach for graphs [18]. Namely, we seek to maximize the posterior probability of $\mathbf{G}$ given partially observed training data $\mathbf{X}_O$ within the space of all DAGs:

$$G_\star = \arg\max_{\mathbf{G} \in \text{DAGs}} p(\mathbf{X}_O|\mathbf{G})p(\mathbf{G}). \tag{1}$$

To optimize over the structure with the DAG constraint in Eq. 1, we resort to recent continuous optimization techniques [25, 63, 64], where a differentiable measure of 'DAG-ness', $\mathcal{R}(\mathbf{G}) = \text{tr}(e^{\mathbf{G} \odot \mathbf{G}}) - D - 1$, was proposed and is zero if and only if $\mathbf{G}$ is a DAG. To leverage this DAG-ness characterisation, we follow Kyono et al. [25], Yu et al. [61] and introduce a regulariser based on $\mathcal{R}(\mathbf{G})$ to favour the DAG-ness of the solution, i.e.

$$G_\star = \arg\max_{\mathbf{G}} \left( p(\mathbf{X}_O|\mathbf{G})p(\mathbf{G}) - \lambda\mathcal{R}(\mathbf{G}) \right). \tag{2}$$

In the following two sections, we present our detailed formulation, training and imputation algorithms of VISL, that allows the model to infer the latent structure $\mathbf{G}$ and impute missing values $\tilde{\mathbf{x}}_U$ in a test sample $\tilde{\mathbf{x}} \in \mathbb{R}^D$ based on the observed $\tilde{\mathbf{x}}_O$.

## 2.2 Generative model and variational inference

For the generation of observation $\mathbf{X}$, we adopt the latent variable model of Fig. 1. Particularly, given an inferred graph $G$ and latent $\mathbf{Z}$, the generative path from $\mathbf{Z}$ to $\mathbf{X}$ is provided in Fig. 1, where we use a graph neural network (GNN) decoder that respects the learned graph structure $G$ and the provided grouping structure. Then the joint model likelihood is

**Algorithm 1** Generative process

$\mathbf{G}_{ij} \sim \text{Bernoulli}(p_{ij})$
**for** $n \in \{1, 2, \cdots, N\}$ **do**
    $\mathbf{Z}_n \sim \mathcal{N}(\mathbf{0}, \sigma_z^2\mathbf{I})$
    $\mathbf{x}_n \sim \mathcal{N}(f_\theta(\mathbf{Z}_n, \mathbf{G}), \sigma_x^2\mathbf{I})$
**end for**

$$p(\mathbf{X}, \mathbf{Z}, \mathbf{G}) = p(\mathbf{G}) \prod_n p(\mathbf{x}_n|\mathbf{Z}_n, \mathbf{G})p(\mathbf{Z}_n). \tag{3}$$

Using this marginal likelihood $p(\mathbf{X})$ is consistent with Bayesian score based causal discovery [18]. As Kaiser and Sipos [22] and Reisach et al. [39] pointed out that other commonly used objectives, e.g. L2 loss in NOTEARS [63], are sensitive to data scaling and result in learning directions towards the high variance nodes. Loh and Bühlmann [28] and Ng et al. [35] showed that using a proper likelihood function can address these problems. Next, we leverage amortized variational inference to sidestep the intractable marginalization of the joint likelihood.

**Amortized variational inference.** The true posterior distribution over $\mathbf{Z}$ and $\mathbf{G}$ in Eq. 3 is intractable since we use a complex deep learning architecture. Therefore, we resort to an efficient amortized variational inference as in Kingma and Welling [24], Kingma et al. [23]. Here, we consider a fully factorized variational distribution $q(\mathbf{Z}, \mathbf{G}|\mathbf{X}) = q_\phi(\mathbf{G}) \prod_{n=1}^N q_\phi(\mathbf{Z}_n|\mathbf{x}_n)$, where $q_\phi(\mathbf{Z}_n|\mathbf{x}_n)$ is a Gaussian whose mean and (diagonal) covariance matrix are given by an *encoder*. For $q(\mathbf{G})$, we consider the product of independent Bernoulli distributions over the edges; that is, the presence of each edge from $i$ to $j$ is associated with a probability $p_{ij}$ to be estimated. With the above formulation, the evidence lower bound (ELBO) is

$$\text{ELBO} = \sum_n \big\{ \mathbb{E}_{q_\phi(\mathbf{Z}_n|\mathbf{x}_n)q(\mathbf{G})}[\log p(\mathbf{x}_n|\mathbf{Z}_n, \mathbf{G}) -$$
$$\text{KL}[q_\phi(\mathbf{Z}_n|\mathbf{x}_n)||p(\mathbf{Z}_n)]] \big\} - \text{KL}[q(\mathbf{G})||p(\mathbf{G})]. \tag{4}$$

Next, we explain our choice of the generator (decoder), which uses a GNN over a learned graph $\mathbf{G}$ to model the interactions between latent variables, representing the information about each group. Then, we focus on the inference network (encoder), representing the mapping from the group of observed variables to its corresponding latent representation.

**Generator**. The generator (i.e., decoder) takes $\mathbf{Z}_n$ and $\mathbf{G}$ as inputs and outputs the reconstructed $\hat{\mathbf{x}}_n = f_\theta(\mathbf{Z}_n, \mathbf{G})$, where $\theta$ are the decoder parameters. In order to respect the pre-defined group structure, as shown in Fig. 1, $\mathbf{Z}_n$ is partitioned into $M$ parts, where $\mathbf{z}_{n,m}$ represents the latent variable for the group of observations $\chi_{n,m}$. This defines a group-wise structured latent space. We adopt a two-step process for the generative path $\mathbf{Z}_n$ to $\mathbf{X}_n$: (i) GNN message passing with respect to the learned graph $\mathbf{G}$ between latent $\mathbf{z}_{n,m}$; (ii) final read-out layer to generate $\mathbf{X}_n$.

**GNN message passing in the generator**. In message passing, the information flows between nodes in $T$ consecutive node-to-edge (n2e) and edge-to-node (e2n) operations [13]. At the $t$-th step, we compute an embedding $\mathbf{h}_{i \to j}^f$ for each edge $i \to j$, called *forward* embedding, which summarizes the

information sent from node $i$ to $j$. Specifically, the n2e/e2n operations in VISL are

$$\text{n2e}: \quad \mathbf{h}_{i \to j}^{(t),f} = \text{MLP}^f \left( \left[ \mathbf{z}_i^{(t-1)}, \mathbf{z}_j^{(t-1)} \right] \right), \tag{5}$$

$$\text{e2n}: \quad \mathbf{z}_i^{(t)} = \text{MLP}^{e2n} \left( \sum_{k \neq i} \mathbf{G}_{ki} \cdot \mathbf{h}_{k \to i}^{(t),f} \right). \tag{6}$$

Here, $t$ refers to the $t$-th iteration of message passing (that is, $\mathbf{Z}^{(0)} = \mathbf{Z}_n$, notice that we omit subindex $n$ for clarity). Finally, $\text{MLP}^f$, and $\text{MLP}^{e2n}$ are MLPs to be trained.

Interestingly, the message passing updates indicate that the information flows between latent nodes if a directed edge is specified in graph $\mathbf{G}$. Hence, the inferred structure $\mathbf{G}$ directly defines relations for latent space $\mathbf{Z}$ which contains the information of pre-defined groups. We show that under certain conditions, the inferred graph $G$ also represents the group-wise structure in observational space, and the corresponding model can be reformulated to a general *structural equation model* (SEM) [37] (see Appx. B).

**Read-out layer in the generator**. After $T$ iterations of GNN message passing, we have $\mathbf{Z}^{(T)}$. Due to we allow $\mathbf{Z}^{(T)}$ and $\mathbf{x}$ to have different dimensions, we apply a final function that maps $\mathbf{Z}^{(T)}$ to the reconstructed $\hat{\mathbf{x}}$, which also respects the pre-defined group structure. Since the observation $\mathbf{x} = [\boldsymbol{\chi}_1, \ldots, \boldsymbol{\chi}_M]$ may contain $\boldsymbol{\chi}_m$ with different dimensions, we adopt $M$ different MLPs, one for each group as the final read-out layer, to respect the group structure. Namely, $\hat{\mathbf{x}} = (g^1(\mathbf{z}_1^T), \ldots, g^M(\mathbf{z}_M^T))^\top$, where $g^m$ represents the MLP for group $m$. Thus, the decoder parameters $\boldsymbol{\theta}$ include the parameters of the following neural networks: $\text{MLP}^f$, $\text{MLP}^{e2n}$ and $g^m$ for $m = 1, \ldots, M$.

**Inference network**. As in standard VAEs, the encoder maps a sample $\mathbf{x}_n$ to its latent representation $\mathbf{Z}_n$. As discussed before, $\mathbf{Z}_n$ is partitioned into $M$ parts, where each $\mathbf{z}_{n,m}$ contains the information of the observation in group $m$. Similar to the read-out layer, we utilize the $M$ MLPs to map groups of observations to the mean/variance of the latent variables:

$$\boldsymbol{\mu}_n = \left( \mu_{\phi_{\mu_1}}^1(\boldsymbol{\chi}_{n,1}), \ldots, \mu_{\phi_{\mu_M}}^M(\boldsymbol{\chi}_{n,M}) \right)^\top, \tag{7}$$

$$\boldsymbol{\sigma}_n = \left( \sigma_{\phi_{\sigma_1}}^1(\boldsymbol{\chi}_{n,1}), \ldots, \sigma_{\phi_{\sigma_M}}^M(\boldsymbol{\chi}_{n,M}) \right)^\top.$$

Here, $\mu_{\phi_{\mu_m}}^m$ and $\sigma_{\phi_{\sigma_m}}^m$ are neural networks for group $m$. When missing values are present, we replace them with a constant as in [34]. Under this formulation, VISL can infer latent variables $\mathbf{Z}_n$ from incomplete $\mathbf{x}_n$ for both training and test data. A graphic representation of how the encoder respects the structure of the latent space is shown in the appendix, Fig. 6(b).

## 2.3 Training VISL

Given the model described above, we propose the training objective to minimize w.r.t. $\theta$, $\phi$ and $\mathbf{G}$:

$$\mathcal{L}_{\text{VISL}}(\theta, \phi, \mathbf{G}) = -\text{ELBO} + \lambda \mathbb{E}_{q(\mathbf{G})} \left[ \mathcal{R}(\mathbf{G}) \right], \tag{8}$$

where ELBO is given by Eq. 4 and the DAG regulariser $\mathcal{R}(\mathbf{G})$ was introduced in Eq. 2 to favor the DAG-ness of learned graph $\mathbf{G}$.

**Evaluating the training loss $\mathcal{L}_{\text{VISL}}$**. VISL can work with any type of data. The log-likelihood term ($\log p_\theta(\mathbf{x}_n|\mathbf{Z}_n, \mathbf{G})$ in Eq. 4) is defined according to the data type. We use a Gaussian likelihood for continuous variables and a Bernoulli likelihood for binary ones. For the inference of $\mathbf{Z}$ and $\mathbf{G}$, the standard reparametrization trick is used to sample $\mathbf{Z}_n$ from the Gaussian $q_\phi(\mathbf{Z}_n|\mathbf{x}_n)$ [23, 24]. To backpropagate the gradients through the discrete variable $\mathbf{G}$, we resort to the Gumbel-softmax trick to sample from $q(\mathbf{G})$ [21, 31]. The $\text{KL}[q_\phi(\mathbf{Z}_n|\mathbf{x}_n)||p(\mathbf{Z}_n)]$ and $\text{KL}[q(\mathbf{G})||p(\mathbf{G}))]$ terms can be obtained in closed-form since they are Gaussian distributions and independent Bernoulli distributions over the edges, respectively. This formulation brings additional advantages in real-life applications since one can easily incorporate domain knowledge and prior information into the VISL framework. For example, if the existence of a specific edge is known a priori, the edge probability can be set to 0/1 in the prior distribution. Finally, the DAG-loss regulariser in Eq. 8 can be computed by evaluating the function $\mathcal{R}$ on a Gumbel-softmax sample from $q(\mathbf{G})$. To adapt the model to different missing levels in the training data $\mathbf{X}$, we adopt the *masking* strategy [15, 30], which drops a random percent

---

**Algorithm 2** Training VISL.

---

**Input** : Training dataset $\mathbf{X}$, possibly with missing values.

**for** *each batch of samples* $\{\mathbf{x}_n\}_{n \in B}$ **do**

> Drop a percentage of the data for each sample $\mathbf{x}_n$.
> Encode $\mathbf{x}_n$ through the reparametrization trick to sample $\mathbf{Z}_n \sim \mathcal{N}(\boldsymbol{\mu}_\phi(\mathbf{x}_n), \boldsymbol{\sigma}_\phi^2(\mathbf{x}_n))$ using Eq.7.
> Use the Gumbel-softmax to sample $\mathbf{G}$ from q($\mathbf{G}$).
> Use decoder to reconstruct $\hat{\mathbf{x}}_n = f_\theta(\mathbf{Z}_n, \mathbf{G})$.
> Calculate the training loss $\mathcal{L}_{\text{VISL}}$ (Eq. 8).
> Gradient step w.r.t. $\phi$ (encoder parameters), $\theta$ (decoder parameters) and $\mathbf{G}$ (posterior edge probabilities).

**Output** : Encoder parameters $\phi$, decoder parameters $\theta$, and posterior probabilities over the edges $\mathbf{G}$.

---

of the observed values during training. The entire training procedure for VISL is summarised in Algorithm 2.

**Two-step training**. After training, we obtain the posterior of the graph $\mathbf{G}$, which respects the underlying structure of the groups as shown in Appx. B. With the trained network, we can impute missing values in the groups where their ancestors contain some observations but if a group has no ancestors no information can be propagated during imputation. After learning the graph structure and to facilitate the imputation task, we introduce a *backwards* edge: for an edge from $j$ to $i$ we denote the backwards edge information as $\mathbf{h}_{i \to j}^b$ which codifies the information that the $i \to j$ edge lets flow from the $j$-th to the $i$-th node. It is defined in the same way as Eq. 5, i.e.,: $\mathbf{h}_{i \to j}^{(t),b} = \text{MLP}^b\left(\left[\mathbf{z}_i^{(t-1)}, \mathbf{z}_j^{(t-1)}\right]\right)$, where $\text{MLP}^b$ is the backward MLP; and the e2n update (Eq.6) is modified to $\mathbf{z}_i^{(t)} = \text{MLP}^{e2n}\left(\sum_{k \neq i} \mathbf{G}_{ki} \cdot \left\{\mathbf{h}_{k \to i}^{(t),f} + \mathbf{h}_{i \to k}^{(t),b}\right\}\right)$.

In summary, we propose a two-stage training process, where the first stage — described in previous sections — focuses on discovering the edge directions between nodes without the $\text{MLP}^b$ (i.e., we do not train the $\text{MLP}^b$). In the second stage, we fix the graph structure $\mathbf{G}$ and continue to train the model with the backward MLP. This two-stage training process allows VISL to leverage the backward MLP for the imputation task without updating the graph structure.

**Revisiting the learning objectives**. The optimal graph of relationships, denoted as $\mathbf{G}_\star$ in Eq. 2, is given by the estimated posterior probabilities of graph $\mathbf{G}$. In addition, the regularizer $\mathcal{R}(\mathbf{G})$ provides a way to evaluate if the resulting graph is a DAG. By tuning the regularizer strength $\lambda$, one can ensure that the resulting $\mathbf{G}^*$ represents a proper DAG.

For imputation, similar to Ma et al. [30], Nazabal et al. [34], the trained model can impute missing values for a test instance $\widetilde{\mathbf{x}}$ as

$$p(\widetilde{\mathbf{x}}_U | \widetilde{\mathbf{x}}_O, \mathbf{X}) = \mathbb{E}_{q_\phi(\mathbf{Z}|\widetilde{\mathbf{x}})q(\mathbf{G})} p(\widetilde{\mathbf{x}}_U | \mathbf{Z}, \mathbf{G}). \tag{9}$$

Therefore, the distribution over $\widetilde{\mathbf{x}}_U$ (missing values) is obtained by applying the encoder and decoder with $\widetilde{\mathbf{x}}$ as input. Previous work [30, 34] also uses variational autoencoders for missing value imputation, which can have problems with overfitting due to spurious correlations [25] even with sufficient training data. One important distinction of VISL is the incorporation of the learned structure $\mathbf{G}$, which helps the model to be robust to spurious correlations.

**Special case: variable-wise relations**. In the above formulation, we have defined VISL for group-wise structure learning. Variable-wise relations can be regarded as a special case. In particular, we can set $M = D$ and $\mathcal{I}_m = \{m\}$ (see Fig. 5 (a) in the appendix), i.e. each group only contains a single variable. Through this modification, we can further simplify the encoder and read-out layer. Instead of using $M$ different MLPs, a single MLP can be shared across all variables since each group has dimension of 1. The mean function for the encoder is then defined as

$$\boldsymbol{\mu}_n = (\mu_\phi(x_{n,1}), \ldots, \mu_\phi(x_{n,D})). \tag{10}$$

One can define encoder variance $\boldsymbol{\sigma}$ (Fig. 6 (a) in the appendix) and the read-out layer $g$ analogously.

**Computational cost.** The main computational bottleneck is the training of VISL, where we require a different inference network and read-out layer for each group $m \in M$. However, some weight sharing schemes can be used to reduce the computational cost. When each group has the same dimensionality, we share the weights of the inference networks. In the case of different dimensionalities, one can

infer the latent variables by parallelizing inference network forward passes to reduce computational cost. Table 11 in appendix provides the wall-clock time comparison for each of the methods for up to 512 variables. We can see that all deep learning based methods (i.e. NOTEARS, DAG-GNN, VISL, etc.) have similar time complexity.

## 3 Related Work

Since VISL simultaneously tackles missing value imputation and structure learning, we review both fields. Moreover, we review recent works regarding improving the deep learning performance with structure learning. Finally, as one of the focused applications of this work is education, we review recent advances of AI in education.

**Structure learning**. Structure learning aims to infer the underlying structures associated with some observations. There are mainly three types of methods: constrained-based, score-based, and hybrid. Constraint-based ones exploit (conditional) independence tests to find the underlying structure, such as PC [47] and Fast Causal Inference (FCI) [48]. They have recently been extended to handle partially observed data through test-wise deletion and adjustments [50, 52]. Score-based methods find the structure by optimizing a proper scoring function. The core difficulty lies in the number of possible graphs growing super-exponentially with the number of nodes [4]. Thus, explicitly solving the optimization can only be done up to a few nodes [7, 36, 46]. Therefore, approximation have been proposed to ease the computational burden, including searching over topological ordering [43, 44, 51], greedy search [3, 38], coordinate descent [1, 11, 17].

Recently, continuous optimization, called *Notears*, has become very popular [63]. *Notears* proposed a differentiable characterization of the DAG to learn the model parameters and graph structures jointly. *Notears* has inspired the development of other methods, *Notears-MLP* and *Notears-Sob* [64], *Grandag* [26], and *DAG-GNN* [61], which extends to model nonlinear relationships between variables. However, their formulations cannot handle missing values and have been observed to be sensitive to data scaling [22]. *DAG-GNN* also adopts a specially-designed GNN to perform structure learning [61]. There are three key distinctions compared to VISL: (i) our model handles the group-wise relationship, while DAG-GNN focuses on variables level; (ii) our model is capable of missing value imputation and group-wise structure learning simultaneously, whereas the original formulation of DAG-GNN and related work require complete data; (iii) VISL adopts Bayesian view for the graphs, compared to a point estimation. VISL is also related to Bayesian DAG learning. Viinikka et al. [55] proposed to sample the graph posterior using MCMC, but it suffers from high computation complexity. VISL, in contrary, can easily be scaled to high dimensional datasets. BCD [6] and DiBS [29] have recently leveraged the approximate inference for scalable Bayesian DAG learning. BCD nets focused on linear Gaussian SEM (v.s. nonlinear relationship by VISL). DiBS adopted a full Bayesian treatment and used a particle sampler to draw from a joint posterior. Compared to VISL, it cannot handle missing values nor variable groups. Geffner et al. [12] recently proposed an end-to-end causal inference framework (DECI) combining causal discovery and inference with missing data. Compared to VISL, DECI is built upon nonlinear additive noise models [19], whereas VISL uses an autoencoder structure with a GNN decoder. When the GNN reaches equilibrium state, Appendix B shows that VISL represents a general SEM, which includes additive noise models as a special case. Apart from the likelihood-based causal discovery, Rolland et al. [40] used score matching to extract causal relationships for non-linear additive noise models. Our work differs in four aspects: (1) structure learning (VISL) v.s. causal discovery; (2) VISL is based on the auto-encoder framework; (3) VISL can handle MAR missing data and (4) VISL uses a Bayesian view of graphs.

**Structure deep learning**. Continuous optimization for learning structures has been used to boost performance in classification. In CASTLE [25], structure learning is introduced as a regulariser for a classification model. This regulariser reconstructs only the most relevant features, leading to improved out-of-sample predictions. In SLAPS [10], the classification objective is supplemented with a self-supervised task that learns a graph of interactions between variables through a GNN. These works focused on leveraging the structure learning instead of advancing its performance.

**Missing values imputation**. The relevance of missing data in real-world problems has motivated a long history of research [9, 41]. A popular approach is to estimate the missing values based on the observed ones through different techniques [45], e.g. Random Forest [49], Bayesian Ridge Regression [2]. Wu et al. [59] explored the use of generative model for missing value imputation,

although fully observed training data is required. This limitation is addressed in both Nazabal et al. [34] with zero-imputing strategy, and Ma et al. [30] with a permutation invariant set encoder. Mattei and Frellsen [32] proposed to use the importance weighted autoencoder, which enables a tighter lower bound than ELBO and leads to improved performance. Ivanov et al. [20] parameterized the imputation as sampling from a conditional distribution, and proposed a method for arbitrary conditioning with VAEs.

**AI in education**. There has been tremendous progress in AI for educational applications, e.g. knowledge tracing [27, 33, 54]; grading students' performance [58]; generating feedback for students working on coding challenges [60]. In particular, most related to VISL is imputing missing values in students' responses. Wang et al. [56] adopts a partial VAE [30] to perform missing value imputation and personalization. However, partial VAE does not consider the structural relations between questions/topics. With the additional insights from structure learning, VISL can provide more information to teachers to help curriculum design than just imputations.

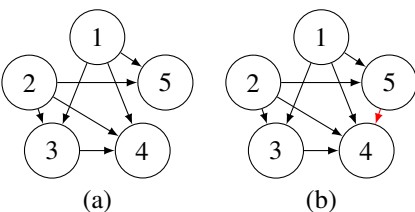

(a)                    (b)

Figure 2: (a): Structure simulated for one of the synthetic datasets with 5 variables. (b): Graph predicted by VISL (when the one on the left is used as the true one). VISL recovers the ground truth graph with one addition edge from 5 to 4.

| | 1 | 2 | 3 | 4 | 5 |
|---|---|---|---|---|---|
| 1 | 0.0 | 0.08 | 0.954 | 0.9021 | 0.943 |
| 2 | 0.069 | 0.0 | 0.949 | 0.922 | 0.953 |
| 3 | 0.084 | 0.092 | 0.0 | 0.638 | 0.343 |
| 4 | 0.073 | 0.077 | 0.195 | 0.0 | 0.219 |
| 5 | 0.085 | 0.094 | 0.446 | 0.517 | 0.0 |

Figure 3: Probability of edges obtained by VISL in the synthetic experiment. By using a 0.5 threshold, we get the predicted graph in Fig. 2(b). Item $(i, j)$ refers to the probability of edge $i \rightarrow j$.

| | RMSE |
|---|---|
| Major | 0.54±0.0032 |
| Mean | 0.22±0.0061 |
| MICE | 0.14±0.0046 |
| Missforest | 0.13±0.0025 |
| PVAE | 0.14±0.0043 |
| VISL | **0.12±0.004** |

Table 1: Imputation results for the synthetic experiment. Mean and standard error over 15 datasets.

| | Adjacency | | | Orientation | | | Causal accuracy |
|---|---|---|---|---|---|---|---|
| | Recall | Precision | $F_1$-score | Recall | Precision | $F_1$-score | |
| PC | 0.42±0.056 | 0.63±0.067 | 0.49±0.056 | 0.22±0.046 | 0.33±0.061 | 0.26±0.051 | 0.33±0.046 |
| GES | 0.45±0.044 | 0.57±0.036 | 0.49±0.038 | 0.25±0.046 | 0.31±0.053 | 0.27±0.049 | 0.36±0.045 |
| NOT. (L) | 0.19±0.028 | 0.44±0.059 | 0.27±0.036 | 0.15±0.023 | 0.37±0.060 | 0.21±0.032 | 0.15±0.023 |
| NOT. (NL) | 0.33±0.039 | 0.49±0.051 | 0.39±0.044 | 0.28±0.032 | 0.42±0.043 | 0.33±0.035 | 0.28±0.032 |
| DAG-GNN | 0.44±0.064 | 0.51±0.062 | 0.46±0.061 | 0.35±0.050 | 0.42±0.052 | 0.37±0.049 | 0.35±0.050 |
| VISL | **0.78±0.084** | **0.73±0.078** | **0.74±0.063** | **0.66±0.13** | **0.60±0.10** | **0.63±0.10** | **0.66±0.13** |

Table 2: Structure discovery results for synthetic experiment (mean and std error over 15 datasets).

# 4   Experiments

We evaluate the performance of VISL in three different problems: a synthetic experiment where the data generation process is controlled, a semi-synthetic problem (simulated data from a real-world problem) (Neuropathic Pain), and the real-world problem that motivated the group-level structure learning (Eedi). The first two datasets are on the variable level. The last one focuses on the group level and real-world usage, and have worked closely with the domain expert to evaluate the results. Additional experiments are presented in the appendix.

**Baselines**. We consider five baselines for the structure discovery task at the variable level. PC [48] and GES [3] are popular constraint-based and score-based approaches, respectively. NOTEARS (NOT.) [63], the non-linear (NL) extension of NOTEARS [64], and DAG-GNN [61] are the other three closely related baselines. Unlike VISL, these baselines cannot deal with missing values. Therefore, we work with complete training data in the first two sections. The Eedi real-world data is only partially observed, where these baselines are not applicable. For the missing data imputation, we also consider five baselines. Mean Imputing (Mean), Majority Vote (Major) (refer to Appx. D for short descriptions), Missforest [49] and MICE [2] are four widely-used imputation algorithms, and PVAE [30] is a recent algorithm based on amortized inference.

|  | Accuracy | AUROC | AUPR |
|---|---|---|---|
| Major | 0.9268±0.0003 | 0.5304±0.0003 | 0.3366±0.0025 |
| Mean | 0.9268±0.0003 | 0.8529±0.0012 | 0.3262±0.0034 |
| MICE | 0.9469±0.0007 | 0.9319±0.0010 | 0.6513±0.0046 |
| Missforest | 0.9305±0.0004 | 0.8915±0.0093 | 0.5227±0.0033 |
| PVAE | 0.9415±0.0003 | 0.9270±0.0007 | 0.5934±0.0046 |
| VISL | **0.9471±0.0006** | **0.9392±0.0008** | **0.6597±0.0053** |

Table 3: Imputation results for neuropathic pain data (mean and std error over five runs).

|  | Adjacency | | | Orientation | | | Causal Accuracy |
|---|---|---|---|---|---|---|---|
|  | Recall | Precision | $F_1$-score | Recall | Precision | $F_1$-score |  |
| PC | 0.046±0.001 | 0.375±0.006 | 0.082±0.001 | 0.024±0.001 | 0.199±0.011 | 0.044±0.002 | 0.058±0.003 |
| GES | 0.110±0.001 | 0.436±0.008 | 0.176±0.002 | 0.082±0.001 | 0.323±0.009 | 0.131±0.003 | 0.121±0.001 |
| NOT. (L) | 0.006±0.000 | 0.011±0.001 | 0.008±0.000 | 0.001±0.000 | 0.001±0.001 | 0.001±0.000 | 0.001±0.000 |
| NOT. (NL) | 0.011±0.001 | **0.644±0.025** | 0.022±0.002 | 0.006±0.001 | 0.354±0.018 | 0.012±0.001 | 0.006±0.001 |
| DAG-GNN | 0.129±0.028 | 0.272±0.101 | 0.128±0.027 | 0.051±0.010 | 0.126±0.059 | 0.050±0.007 | 0.051±0.010 |
| VISL | **0.261±0.006** | 0.637±0.009 | **0.370±0.005** | **0.236±0.007** | **0.573±0.005** | **0.334±0.006** | **0.245±0.006** |

Table 4: Structure discovery results for neuropathic pain data (mean and std error over five runs).

**Metrics**. Imputation performance is evaluated with standard metrics such as RMSE (continuous variables) and accuracy (binary variables). For binary variables , we also provide the area under the ROC and the Precision-Recall curves (AUROC and AUPR, respectively), which are especially useful for imbalanced data (such as Neuropathic Pain). We follow common practice [14, 52] regarding structure discovery performance, and consider metrics on the *adjacency* and *orientation*. While the former does not take into account the direction of the edges, the latter does. We report recall, precision and $F_1$-score. We also provide *causal accuracy*, a discovery metric that considers orientation [5].

## 4.1 Synthetic experiment

We simulate fifteen synthetic datasets. For each simulated dataset, we first sample the true structure **G**; see Fig. 2(a) for an example. The appendix provides detailed generation mechanism, including a visualisation of the data in Fig. 7. For each dataset, we simulate 5000 training and 1000 test samples.

**Imputation performance**. VISL outperforms the baselines in terms of imputation across all synthetic datasets ( Table 1).The results grouped by the number of variables are presented by Table 8 in the appendix.

**Structure discovery performance**. VISL obtains better performance than the baselines, see Table 2. The results split by the number of variables are shown in the appendix, Table 10. Specifically, we observe VISL consistently outperforms the baseline method in all metrics considered with all datasets (see Table 10). In this small synthetic experiment, it is possible to visualize the predicted graph. Fig. 3 shows the posterior probability of each edge (i.e. the estimated matrix **G**) for the simulated dataset in Fig. 2(a). Using a threshold of 0.5, we obtain the predicted graph in Fig. 2(b). We observed that VISL recovers the ground truth graphs with one additional edge. The sources of its advanced performances are twofold: 1) VISL uses the ELBO (Eq. 4) as training objective, a surrogate for marginal likelihood. Kaiser and Sipos [22], Reisach et al. [39] reported that L2 loss used in NOTEARS is sensitive to data scaling and directs the learning of directions towards high variance nodes. This can be resolved by using a proper likelihood training objective [28, 35]; 2) instead of using observed variables to construct the structural equation model (NOTEARS) or designing a specialized GNN structure (DAG-GNN), VISL is more flexible in terms of transforming the observed variables into a latent distribution via the encoder, and adopting a general message-passing GNN decoder.

Finally, VISL can scale to large data in terms of data points and dimensionality. We demonstrate the computational efficiency with synthetic data ranging from 4 to 512 nodes in the appendix, Table 11.

## 4.2 Neuropathic pain dataset

We evaluate our method using a benchmark in healthcare applications [53]. The dataset contains records of patients regarding the symptoms associated with neuropathic pain. There are 222 variables

|  | Accuracy | AUROC | AUPR |
|---|---|---|---|
| Major | 0.6260±0.0000 | 0.6208±0.0000 | 0.7465±0.0000 |
| Mean | 0.6260±0.0000 | 0.6753±0.0000 | 0.6906±0.0000 |
| MICE | 0.6794±0.0005 | 0.7453±0.0007 | 0.7483±0.0010 |
| Missforest | 0.6849±0.0005 | 0.7219±0.0007 | 0.7478±0.0008 |
| PVAE | 0.7138±0.0005 | **0.7852±0.0001** | **0.8204±0.0002** |
| VISL | **0.7147±0.0007** | 0.7815±0.0008 | 0.8179±0.0006 |

Table 5: Imputation results for Eedi topics dataset (mean and standard error over five runs).

|  | Adjacency | | Orientation | |
|---|---|---|---|---|
|  | Expt 1 | Expt 2 | Expt 1 | Expt 2 |
| *Random* | 2.04 | 2.08 | 1.44 | 1.40 |
| DAG-GNN | 2.04 | 2.32 | 1.68 | 1.68 |
| VISL | **3.60** | **3.70** | **2.76** | **2.60** |

Table 6: Average expert evaluation of the topic relationships. Cohen's $\kappa$ inter-annotator agreement is 0.72 for adjacency and 0.76 for orientation (substantial agreement).

in this dataset. Unlike the previous experiment with continuous data, this dataset has binary variables indicating the symptoms. The train and test sets have 1000 and 500 patients, respectively.

**Imputation performance**. VISL shows competitive or superior performance when compared to the baselines, see Table 3. Notice that AUROC and AUPR allow for an appropriate threshold-free assessment in this imbalanced scenario. Indeed, as expected from medical data, the minority of values are 1 (symptoms); here, the prevalence of symptoms is around 8% in the test set. Interestingly, it is precisely in AUPR where the differences between VISL and the rest of the baselines are larger except MICE, whose performance is very similar to VISL in this dataset.

**Structure discovery results**. As in the synthetic experiment, VISL outperforms the causality-based baselines; see Table 4. Notice that NOTEARS (NL) is slightly better in terms of adjacency-precision, i.e. the edges that it predicts are slightly more reliable. However, this is at the expense of a significantly lower capacity to detect true edges, see the recall and the trade-off between both ($F_1$-score).

### 4.3 Eedi topics dataset

Finally, we evaluate our method on an even more challenging real-world dataset in education to discover the group-wise structure between topics while the observations are question-answer pairs under these topics. This is an important real-world problem in the field of AI-powered educational systems [56, 57]. This dataset is very sparse, with 74.1% of the values missing. The dataset contains the responses from 6147 students to 948 mathematics questions. The 948 variables are binary (1: correct and 0 otherwise). These questions target specific mathematical concepts and are grouped into a meaningful hierarchy of *topics*; see Fig. 4 in Appx. E. Here we apply VISL to find the structures among the topics using the third level hierarchy (Fig. 4), resulting in 57 group-level nodes.

**Imputation results.** VISL achieves competitive or superior performance when compared to the baselines (Table 5). Although the dataset is more balanced (54% of the values are 1), we provide AUROC and AUPR for completeness. Notice that this setting is more challenging since the information flows at less granular level (i.e. group). Interestingly, even in this case, VISL obtains similar or improved imputation results compared to the baselines.

**Structure discovery results between groups.** Most of the baselines used so far cannot be applied here because i) they cannot deal with missing data or ii) they cannot learn group-level relationships. DAG-GNN is the only one that can be adapted to satisfy both properties. For missing data, we adapt DAG-GNN following the same strategy as in VISL, i.e. replacing missing entries with a constant value. For the second one, we further adapt it by using group-specific mappings like VISL to cope with arbitrary groups. We also include another baseline, *Random*, where the structures between topics are randomly sampled from a Bernoulli distribution. Due to the lack of ground truth relationships, we ask two experts (teachers) to assess the validity of the relationships found by VISL, DAG-GNN, and *Random*. For each relationship, they are asked to rate a value from 1 (strongly disagree) to 5 (strongly agree) for the adjacency (whether it is sensible to connect the two topics) and the orientation (whether the first one is a prerequisite for the second one). We release the the complete list of relationships and expert evaluations for VISL, DAG-GNN, and *Random*; see Table 15, Table 16, and Table 17, respectively. Thus, the results is reproducible and allow the community to build on this data as a structure discovery benchmark. In summary, Table 6 shows here the average evaluations: we see that the relationships discovered by VISL score much more highly across both metrics than the baseline models.

Another interesting aspect is how the relationships between level-3 topics are distributed in higher-level topics. Intuitively, it is expected that most of the relationships happen *inside* higher-level topics (e.g. Number-related concepts are more probably related to each other than to Geometry-related ones). Table 7 in appendix shows the distribution for the compared methods. Indeed, notice that the percentage of inside-topic relationships is higher for VISL (82%) and DAG-GNN (42%) than for *Random* (34%). An analogous analysis for the 25 level-2 topics is provided in the appendix; see Table 12 (VISL), Table 13 (DAG-GNN), and Table 14 (*Random*). 6% of the connections happen inside level 2 topics for *Random*, it is 14% for DAG-GNN and 36% for VISL.

**Education Impact.** Lastly, to make a real-world impact, we have been provided with an additional education dataset in the same format as Eedi by an education organization to help provide insight for math curriculum building. The final structure among all topics found by VISL is presented by figure Fig. 8 in the appendix. This figure provides insights into which topics are foundational and need to be covered earlier (topics with many originating edges), or later (topics with many incoming edges). This allowed us to re-evaluate the order of topics in a nationwide used secondary curriculum. Specifically, topics such as "arithmetic" or "properties of shapes" were moved earlier in the curriculum, while topics such as "negative numbers" or "proportion and similarity" were moved to a later stage in the curriculum. Another interesting example found by the domain expert are "Venn diagrams", which were originally taught in year 9/10 and are now suggested to be moved to year 7. Experts found that the topic "Venn diagram" has been a useful tool in teaching other topics which are currently taught before year 10. This emphasises the real-world impact our model can have in planning curricula.

## 5    Conclusions

We introduced VISL, a novel approach that simultaneously performs group-wise structure discovery and learns to impute missing values. Both tasks are performed jointly: imputation is informed by the discovered relationships and vice-versa, leading to improved performance for both tasks. Moreover, motivated by a real-world problem, VISLshows its impact in the real-world education domain to aid domain experts in setting up curriculum. Despite of the improved performance on structure learning with missing data, VISL can be further extended in several directions. First, one potential limitation is that the inferred structure from VISL is not causal. Appendix B shows that VISL satisfies the cauasl Markov assumption under equilibrium, which opens a door for potential causal claims. Another direction for future work is to extend the M(C)AR assumption to *missing not at random*(MNAR), which can be more practical for real world usage. In the end, the performance can further be boosted by designing better graph posteriors beyond an independent Bernoulli distribution.

## Acknowledgments and Disclosure of Funding

PMA acknowledges project PY20_00286 funded by Junta de Andalucía.

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
