# OpenReview forum: "Simultaneous Missing Value Imputation and Structure Learning with Groups"
_NeurIPS.cc/2022/Conference — NeurIPS 2022 Accept_

### Official Review · Reviewer_9NtP · 2022-07-11

**Rating:** 7
**Confidence:** 3
**Soundness:** 3 good
**Presentation:** 3 good
**Contribution:** 3 good

**Summary:**

This work introduces VISL (missing value imputation with structural learning) which simultaneously should learn relations between groups of variables and do missing value imputation. The model consists of a Variational Autoencoder with a structure imposed on the latent space. Each group of variables gets its own latent space and the specific relation between the subspaces is treated as a latent variable itself. A GNN generator works on the latent variables to reconstruct the missing values.

**Questions:**

*Q1*:
I see some hints at interpreting the learnt structure causally, e.g.
"However, it is commonly impossible to perform randomized controlled trials for many real-world applications due to ethical or cost considerations.".
Can the learnt structure be interpreted causally, without any further assumptions? Could you be more clear about whether you are interpreting the structure causally and if so what allows you to do that?

*Q2*:
"where we assume the data are missing completely at random (MCAR) or missing at random (MAR)"
Do you have any notion whether your specific dataset is truly M(C)AR or if it could be MNAR? Could the missing mechanism depend on the missing values themselves? How would such a situation affect the learnt structure and the imputations?

*Q3*:
"To adapt the model to different missing levels in the training data X, we adopt the masking strategy [14, 27], which drops a random percent of the observed values during training." Does the model then need fully observed data for training?

*Q4*:
"it incorporates the learned structure G into the imputation, which helps the model avoid over-fitting due to spurious correlations [22]". Is that really the case? Couldn't the model in a small data regime just overfit G as well?

Regarding VAEs for missing data, perhaps you would find [1, 2] interesting as well.

[1] Mattei, Pierre-Alexandre, and Jes Frellsen. "MIWAE: Deep generative modelling and imputation of incomplete data sets." International conference on machine learning. PMLR, 2019.
[2] Ivanov, Oleg, Michael Figurnov, and Dmitry Vetrov. "Variational autoencoder with arbitrary conditioning." arXiv preprint arXiv:1806.02382 (2018).


**Limitations:**

The authors have addressed limitations

**Strengths And Weaknesses:**

*Originality*:
This work seems to connect density learning using VAEs with structure learning in the latent space in a novel way to do both structure learning and missing value imputation.

*Quality*:
The model and experimental setup is well described and most claims are backed up.

*Clarity*:
I think the paper is well written and fairly well structured.

*Significance*:
The paper provides a way of handling missing values in structure learning, which is valuable in most real-life settings.

---

> ### Author Response · Authors · 2022-08-01
> **Author's initial response**
>
> ### Is this causal discovery
> No, we do not claim this is a causal discovery method. Thus, we are very cautious about using the word 'causal' in our paper. We removed the sentence that causes confusion in the revised paper. However, appendix B raises an interesting theoretical insight: when the GNN decoder reaches the equilibrium states with the invertible read-out layer, the joint distribution satisfies the causal Markov condition. This condition is a fundamental requirement for all causal discovery methods. Therefore, this insight opens a door for future research about VISL-inspired causal discovery methods.
>
> ### Is the dataset MAR or MNAR
> For the synthetic dataset, we manually create an MCAR missing pattern. However, for the real dataset, testing whether it is MAR or MNAR is a challenging problem. Typically, a case-by-case analysis is required. We want to point out that how to handle MNAR is a key research question and almost all existing approaches require additional assumptions of the MNAR case. Those additional assumptions themselves are difficult to test for the real-world dataset.
>
> If the missing mechanism is MNAR rather than MAR/MCAR, the resulting VISL will be biased. From Appendix A, we cannot ignore the missing mechanism in the objective function under MNAR. Otherwise, the learning signal will be biased. We add this discussion in appendix A in the revised paper.
>
> ### Require complete training data
> Under our formulation, VISL does not require complete training data. In fact, VISL can handle arbitrary missing proportions in the training data. This is also highlighted in algorithm 2 of the revised paper. To be specific, when encountering an incomplete entry $x_n$ during training, from algorithm 2, we first manually mask some of the entries called $x_m$. Therefore, $x_n$ now consists of $x_u$ (missing value), $x_m$ (masked value) and $x_o$ (observed value after masking). Then $x_o$ will be transformed to latent variable $Z_n$ through the inference network, where $x_u$, $x_m$ are replaced with a constant value (0 in our paper). Then, $Z_n$ will be transformed through the GNN decoder and read-out layer with graph $G$, and output reconstructed $\bar{x}_o$, $\bar{x}_m$ and $\bar{x}_u$. The reconstruction loss of ELBO will be computed for $\bar{x}_o$ and $\bar{x}_m$. Thus, through masking, VISL can learn to impute the missing values from the observed ones.
>
> Although VISL can handle arbitrary missingness, the imputation performance crucially depends on the missing mechanism and proportions. In an extreme case, if a variable is completely missing from the training data, no method can impute it based on the observations. Thus, for VISL to work well, we require at least some observations for each variable in the training data.
>
> ### Overfitting due to spurious correlation
> Yes, if the dataset is very small overfitting can potentially happen. The main point we are trying to make is that spurious correlation can lead to overfitting even with a large amount of data. We have revised the sentence.
>
> However, even with small training data, the Bayesian view of VISL should help to prevent overfitting. The ELBO term consists of an entropy encouragement term of the graph posterior. Thus, when the dataset is small, this term will dominate, leading to diversified graphs. Hence, the function parameters should not overfit the data with diversified graphs.
>
> ### Relevant papers
> We have added them in the related work section.

---

### Official Review · Reviewer_7oh7 · 2022-07-13

**Rating:** 5
**Confidence:** 3
**Soundness:** 3 good
**Presentation:** 2 fair
**Contribution:** 2 fair

**Summary:**

In this paper, the authors propose VISL, a structure learning method that simultaneously infers structures between groups of variables under missing data and performs missing value imputations.  The authors conduct extensive experiments on synthetic, semi-synthetic, and real-world education data sets and they show improved performances on both imputation and structure learning.


**Questions:**

The authors could add discussion and comparison with score matching based method [1].

[1] Rolland, Paul, et al. "Score matching enables causal discovery of nonlinear additive noise models." arXiv preprint arXiv:2203.04413 (2022).




**Limitations:**

As mentioned above, the authors could present additional theoretical results on structure learning under missing values to make the paper stronger. Theoretical study on how missing values will affect the performance of structure learning could technically strengthen the paper.

**Strengths And Weaknesses:**

Strengths:
	1	The writing of the paper is good. Technical sections are easy to follow.
	3	Extensive experiments have been conducted to validate the method.

Weakness:
	1. The contribution and novelty of the paper are limited.  The model proposed in this paper basically combines existing techniques, i.e. structure learning regularization (NOTEARS) and missing value imputation, and uses variational inference for model parameter learning.

        2. Technical depth of the paper is insufficient. The authors could present additional theoretical results on structure learning under missing values to make the paper stronger. Theoretical study on how missing values will affect the performance of structure learning could technically strengthen the paper.

---

> ### Author Response · Authors · 2022-08-01
> **Author's initial response**
>
> ### Technical contribution is not sufficient
> First, we want to highlight that our claimed contribution is to simultaneously tackle group-wise structure learning when missing data is present. Advancing the theoretical understanding of how missing data affects structure learning is **not the goal of this paper**. In fact, this topic alone would be worth a separate paper. Instead, our submission is more targeted toward providing a methodology to tackle a real-world problem. The main contributions are
> 1. The messaging passing GNN decoder with two-stage training and forward, backward MLPs for imputation.
> 2. Group-wise inference network that can handle incomplete inputs.
> 3. We conduct extensive empirical evaluations, including on real-world datasets, to confirm the effectiveness of VISL. For the education application, the results from VISL have been verified by the domain experts (highlighted in appendix E of the revised paper) for real-world impact. The above empirical evidences support our claimed contributions, which is also confirmed by the other reviewers: `It addresses an important and well-defined problem.` (Reviewer bhzA) and `The paper provides a way of handling missing values in structure learning, which is valuable in most real-life settings.`(Reviewer 9NtP)
>
> Although our paper focuses on providing a new methodology, we additionally provide a theoretical justification on why VISL can handle MAR and MCAR in Appendix A. Also, we provide another theoretical insight on the potential connections of VISL to the causal Markov assumption in Appendix B, which opens a door for future research on VISL-based causal discovery. Thus, we believe the methodology, extensive empirical evaluation and theoretical insights together account for reasonable contributions.
>
> ### Add discussion about score matching for causal discovery
> We have added this in the related work section. However, we do not think this can be directly compared to VISL, since they are significantly different in four aspects: (1) VISL is a structure learning method instead of a causal discovery approach, and no causal claims have been made in our paper; (2) VISL is based on an autoencoder structure, whereas Rolland et al. is based on an additive noise model; (3) one advantage of VISL is to handle the MAR missing values, whereas Rolland et al. requires complete data; (4) VISL adopts a Bayesian view of structure learning whereas Rolland et al. extracts a single graph from the score matching objective.

---

> > ### Author Response · Authors · 2022-08-08
> > **Looking forward to hearing your feedback**
> >
> > Dear reviewer 7oh7
> >
> > We would like to thank you for spending your time carefully evaluating our submission and providing valuable feedback.
> >
> > As the discussion period is expected to conclude soon, we hope our response addresses your concerns. We look forward to hearing your feedback if you have any further questions. We are happy to engage in further discussions.
> >
> > Cheers,
> >
> > Author

---

> > > ### Comment · Reviewer_7oh7 · 2022-08-08
> > > **Response to authors' response**
> > >
> > > The authors addressed some issues.
> > >
> > > However, the technical novelty and the contributions of this paper are still insignificant. Message passing with GNNs, and missing value imputation with structure learning are sort of well-known technics and not hard to address and implement.
> > >
> > > I will add 1 to the score.

---

### Official Review · Reviewer_bhzA · 2022-07-15

**Rating:** 6
**Confidence:** 3
**Soundness:** 4 excellent
**Presentation:** 3 good
**Contribution:** 3 good

**Summary:**

This paper proposes a new approach for learning the structural dependencies between variables in the latent space that can also be used for imputing the missing values.

**Questions:**

* How to decide the number of groups $M$ when we do not know it in advance?
* What if the missing values are also present in the training data?
* How do the authors explain the huge difference between the performance of the proposed method and the baseline methods on the simulated data (Table 2)? Maybe an unfair simulation that favors the proposed method?
* What is the computational complexity of the proposed method? How does it compare with baseline approaches?
* What are the limitations of the proposed method?

**Limitations:**

The limitations of the proposed method are not discussed within the text and would be nice if the authors can add it to the revised version.

**Strengths And Weaknesses:**

*** Strengths***
* The text is well-written and very clear.
* It addresses an important and well-defined problem.
* The experimental results are compelling.

*** Weaknesses***
* The number of groups in data ($M$) is an essential hyperparameter for the proposed model. However, the authors provide no guideline for deciding it (within optimization? Model selection? Prior knowledge?). In a certain application, what if we do not know $M$ in advance?
* It seems that we need to have access to complete data (without missing values) at the model training stage. This can be very restrictive in real applications. What if the missing values are also present in the training data?

---

> ### Author Response · Authors · 2022-08-01
> **Author's initial response 1/2**
>
> ### How to choose and define the hyperparameter $M$?
> In this paper, we assume that $M$ is provided as priori information. In our experiments, questions are naturally grouped into topics and $M$ is the number of topics. When $M$ is not available, this relates to casual representation learning which aims to represent low-level signals into groups. This is out of the scope of this work. We have clarified that M is required in section 2 and added a discussion to the future work.
>
> ### Does VISL require complete training data?
> Under our formulation, VISL does not require complete training data. In fact, VISL can handle arbitrary proportions of missingness in the training data. This is also highlighted in algorithm 2 of the revised paper. To be specific, when encountering an incomplete entry $x_n$ during training, from algorithm 2, we first manually mask some of the entries called $x_m$. Therefore, $x_n$ now consists of $x_u$ (missing value), $x_m$ (masked value) and $x_o$ (observed value after masking). Then $x_o$ will be transformed to the latent variable $Z_n$ through the inference network, where $x_u$, $x_m$ are replaced with a constant value (0 in our paper). Then, $Z_n$ will be transformed through the GNN decoder and read-out layer with graph $G$, and output reconstructed $\bar{x}_o$, $\bar{x}_m$ and $\bar{x}_u$. The reconstruction loss of ELBO will be computed for $\bar{x}_o$ and $\bar{x}_m$. Thus, through masking, VISL can learn to impute the missing values from the observed ones.
>
> Although VISL can handle arbitrary missingness, the imputation performance crucially depends on the missingness mechanism and proportions. In an extreme case, if a variable is completely missing from the training data, no method can impute it based on the observations. Thus, for VISL to work well, we require at least some observations for each variable in the training data.
>
> ### Huge performance difference in Table 2
> We believe there are two advantages of VISL compared to baselines: (1) the VISL objective is based on variational inference, a surrogate for maximum likelihood. Kaiser, (2021) and Reisach, (2021) pointed out that the L2 loss in NoTears is sensitive to data scaling and directs learning in the directions of high variance nodes. Loh, (2014) and Ng, (2020) showed that a proper likelihood function (e.g. Gaussian likelihood in GOLEM and VISL) can address this issue. (2) NoTears directly uses the observed variables to build the structural equation model. Instead, VISL first transforms the variables to a latent space via the inference net. This maps the scalar variables to a distribution over latent vectors and improves the model flexibility. One should note that although the hidden layer output in NoTears MLP also maps scalar variables to vectors, the latent vectors in VISL represent samples from the latent distribution. We also highlight the above discussion in the experiment section.
>
> As for the fair comparison, we have included a detailed explanation of how the baselines are implemented. We highlighted it in the revised version. In a nutshell, all baselines are implemented by either trusted repositories (PC and GES) or by the original author (NOTEARS, DAG-GNN, etc.) with their recommended hyperparameters.
>
> Another possible reason is that the synthetic data only has $5$ variables with sparse edge connections, which means incorrectly removing or adding an edge can lead to a large decrease in the F1 score.
>
> ### Computational Complexity
> In the revised paper, we have added appendix C to discuss the computational cost of VISL. To summarize, VISL has similar computational complexity compared to other GNN and continuous optimization based methods.
> The main computational bottleneck is the requirement of multiple neural networks for each group. However, the weight-sharing trick and parallelization can be used to reduce computational costs. More importantly, we highlighted Table 11 in the appendix, where we report the actual wall-clock time comparison between different baselines with dimensions up to 512. The overall time consumption of VISL is similar to other continuous optimization based methods.

---

> > ### Author Response · Authors · 2022-08-01
> > **Author's initial response 2/2**
> >
> > ### Limitations
> > We add a section in the appendix to discuss the potential limitations and future research directions for VISL.
> >
> > Specifically, the first limitation is that although VISL is a structure learning method, we cannot claim the inferred structure is causal. However, in appendix B, we showed that when the GNN decoder reaches an equilibrium state with the invertible read-out layer, the learned joint distribution satisfies the causal Markov conditions, which is one of the fundamental assumptions for causal discovery. Thus, this opens a door for investigating what assumptions and modifications are required to make VISL a valid causal discovery approach.
> >
> > Another limitation of VISL is the assumption of the missing mechanism. Currently, VISL can only handle MCAR or MAR but not MNAR. From Appendix A, an MNAR mechanism can lead to biased model parameters and incorrect graph posterior.
> >
> > Another potential future research is to design better variational distribution for graph posteriors and perform joint inference for both graph and function parameters. Currently, the independent Bernoulli distribution cannot incorporate edge co-dependencies of the graph and the point estimate of function parameters is not graph-dependent.
> >
> > In the end, if one does not have predefined information about group $M$, causal representation learning can be a useful candidate for extracting such information. Thus, how to combine VISL with causal representation learning is another interesting research direction.
> >
> > ---
> > - Kaiser M, Sipos M. Unsuitability of NOTEARS for causal graph discovery[J]. arXiv preprint arXiv:2104.05441, 2021.
> >
> >
> > - Reisach A, Seiler C, Weichwald S. Beware of the simulated dag! causal discovery benchmarks may be easy to game[J]. Advances in Neural Information Processing Systems, 2021, 34: 27772-27784.
> >
> >
> > - Loh P L, Bühlmann P. High-dimensional learning of linear causal networks via inverse covariance estimation[J]. The Journal of Machine Learning Research, 2014, 15(1): 3065-3105.
> >
> >
> > - Ng I, Ghassami A E, Zhang K. On the role of sparsity and dag constraints for learning linear dags[J]. Advances in Neural Information Processing Systems, 2020, 33: 17943-17954.

---

### Author Response · Authors · 2022-08-01
**Author's revised submission**

Dear reviewers

Thank you all for the valuable feedback regarding our submission.
We have revised both the main text and supplementary material according to our initial responses. In our revision, we use the tag **highlight** and **new** together with the reviewers' name codes to indicate the key information and newly added materials regarding their questions, respectively.

Hope this can clarify the raised issues.

Best,

Paper 5969 author

---

### Meta-Review · Area_Chair_P7LY · 2022-08-27

**Recommendation:** Accept
**Confidence:** Less certain

**Metareview:**

In this paper, the authors propose VISL, a structure learning method that simultaneously infers structures between groups of variables under missing data and performs missing value imputations. The authors conduct extensive experiments on synthetic, semi-synthetic, and real-world education data sets and they show improved performances on both imputation and structure learning.

The reviewers overall agree that this is a strong and acceptable contribution. 7oh7 has some remaining concerns about the novelty of the proposed approach as the individual components used in the approach "are sort of well-known technics and not hard to address and implement". I nevertheless think that the proposed approach is a worthwhile and powerful combination for Missing Value Imputation. Two reviewers bhzA and 9NtP are fairly confident to accept the manuscript.

Overall, this work is an important step towards better missing value imputation in more challenging settings and I support the acceptance of the manuscript.

**Award:**

No

---

### Decision · Program_Chairs · 2022-09-14

Accept